# Difficulty in Emotion Regulation and Self-Concealment as Mediators of the Link Between Psychological Distress and Disordered Eating Behavior in Emerging Adult Women

**DOI:** 10.3390/bs15091259

**Published:** 2025-09-15

**Authors:** Duckhyun Jo, Mary L. Hill, Akihiko Masuda

**Affiliations:** 1Department of Psychology, University of Houston, Houston, TX 77204, USA; 2Department of Psychiatry, University of North Carolina, Chapel Hill, NC 27514, USA; mary_hill@med.unc.edu; 3Department of Psychology, University of Hawaiʻi at Mānoa, Honolulu, HI 96822, USA; amasuda4@hawaii.edu

**Keywords:** eating behavior, emotion dysregulation, self-concealment, structural equation modeling

## Abstract

Objective: Emerging adulthood often involves a greater degree of disordered eating behaviors, especially among women. In this psychosocial context, extant evidence suggests that psychological distress is a major contributing factor to disordered eating behaviors. The present cross-sectional study examined whether psychological distress was positively associated with disordered eating behavior in a sample of emerging adult women, and whether difficulty in emotion regulation, self-concealment, or both together, at least partially explained this association. Method: Participants were 723 emerging adult women aged 18 to 25 years old (*M*_age_ = 19.5, *SD*_age_ = 1.6) who were recruited from a four-year public university in Hawaii, USA. Upon the completion of the informed consent procedure, they voluntarily completed an online survey package that included the self-report measures assessing disordered eating behavior, psychological distress, difficulties in emotion regulation, and self-concealment. Results: We found that psychological distress was positively associated with disordered eating behaviors. We also found that both difficulty in emotion regulation and self-concealment partially accounted for the positive association between psychological distress and disordered eating behavior. Discussion: Future research should examine the conceptual and applied implications of these findings further.

## 1. Introduction

Emerging adulthood, a developmental period spanning from ages 18 to 25 years old ([2]), often involves elevated disordered eating behaviors, especially among women (e.g., [3]; [49]). In Westernized cultures, disordered eating behaviors often occur in secret and function as a way of coping with difficult thoughts and emotions, such as shame ([18]). Disordered eating behaviors that emerging adult women commonly endorse include restricting food intake, binge eating, and purging ([21]; [28], [27]; [41]). In recent years, researchers have examined psychosocial factors that contribute to disordered eating behaviors in this age group (e.g., [16]), including psychological distress ([34]; [47]), difficulty in emotion regulation ([19]), and self-concealment ([31]).

### 1.1. Psychological Distress and Disordered Eating Behaviors

The link between psychological distress and disordered eating behavior has been integrated into conceptual and applied models of disordered eating concerns for years (e.g., [56]). For example, psychological distress in response to difficult life events has been theorized as a major contributing factor for the development and maintenance of disordered eating behaviors within traditional and contemporary cognitive behavioral theories (see [15]; [24]; [58]). As noted earlier, emerging adulthood is a particularly vulnerable period during which women are at greater risk for disordered eating. Existing research with non-clinical samples of college women and men together ([29]; [41]) and undergraduate college women ([47]) has reported positive associations between psychological distress and disordered eating behavior.

### 1.2. Difficulty in Emotion Regulation and Self-Concealment as Underlying Mechanisms

Cognitive and behavioral theories of disordered eating concerns emphasize the role of emotion regulation in the development and maintenance of disordered eating behaviors, especially in the context of greater psychological distress ([15]; [24]). These theories postulate that the positive association between psychological distress and disordered eating behaviors is established in part through difficulty in emotion regulation (e.g., [14]). Difficulty in emotion regulation often refers to one’s inability to recognize or manage negative emotions and psychological distress ([19]) and may include the lack of acceptance of emotional responses, difficulty performing purposeful behavior, difficulty controlling impulses, limited access to strategies for effectively coping with emotions, and difficulty identifying and differentiating emotions ([53]). Difficulty in emotion regulation is a transdiagnostic process across a range of eating disorder symptoms and weight concerns ([8]; [20]; [30]).

Self-concealment is another construct that reflects an underlying maladaptive regulation process in the context of disordered eating ([11]; [32]; [37]; [39]). In applied psychology literature, self-concealment is a general and stable regulation tendency to keep distressing and potentially embarrassing personal information from others ([12]; [31]), and it involves the processes of (a) possessing a distressing personal secret, (b) keeping it from others, and (c) avoiding or feeling apprehensive about self-disclosure ([31]). The process of self-concealment seems to be manifested in both intrapersonal and interpersonal contexts ([32]). Self-concealment is positively associated with both psychological distress ([44], [45]) and disordered eating ([40]; [39]) in young adults (e.g., college women and men).

Finally, difficulty in emotion regulation and self-concealment are interrelated yet distinct processes implicated in disordered eating behavior ([44], [45]). Difficulty in emotion regulation reflects an intrapersonal struggle to manage psychological distress ([19]), while self-concealment involves maladaptive, secretive regulation across intrapersonal and interpersonal contexts ([32]). Further research should explore how these constructs independently and jointly account for the relationship between psychological distress and disordered eating behavior in emerging adult women.

### 1.3. Present Study

In this cross-sectional study of emerging adult women, we first examined the positive association between psychological distress and disordered eating behavior. We then tested whether the positive association between psychological distress and disordered eating behavior was established indirectly through difficulty in emotion regulation, self-concealment, or both. We hypothesized that psychological distress would predict disordered eating behavior, and that this relationship would be at least partially explained by emotion regulation difficulties and self-concealment.

## 2. Method

### 2.1. Participants

Data were obtained through an online survey administered to 1402 undergraduate students enrolled at a four-year public university in Hawaiʻi, USA. To promote data integrity, the survey included four embedded attention-check items (e.g., “please select option 4 for this item”) distributed evenly throughout the questionnaire. The inclusion criteria were (a) participants who self-identified as women, (b) between 18 and 25 years of age, and (c) fluent in English. Participants presenting with psychotic symptoms were excluded from the study. Based on these parameters, 679 individuals were excluded, resulting in a final analytic sample of 723 women within the specified age range.

### 2.2. Procedure

This research was approved by the Institutional Review Board affiliated with the senior author’s university. Recruitment took place within psychology courses (e.g., Introduction to Psychology, Research Methods, Developmental Psychology, and Abnormal Psychology) between March 2018 and December 2020. Data collection was conducted through a secure online platform (Sona and Qualtrics), where participants completed the survey anonymously in exchange for course credit. Prior to participation, individuals received information outlining the study’s purpose and procedures and provided informed consent.

### 2.3. Measures

#### 2.3.1. Demographic Form

A demographic questionnaire was used to capture variables central to the study, including participants’ age, self-identified gender, racial/ethnic background, and sexual identity, alongside other relevant demographic details. Age was treated as a continuous variable, while gender identity was categorized as “man,” “woman,” or “other.” Racial and ethnic identification was classified into nine groups: Native American, Latinx, Asian, Pacific Islander, White, Black, Hawaiian, Other, and Bi-racial/Multicultural (i.e., multiracial). Sexual identity was coded using four categories: heterosexual, homosexual, bisexual, and other.

#### 2.3.2. Disordered Eating Behavior

The Eating Attitudes Test-26 Behavior Subscale (EAT-26-B; [17]; [41]) comprises nine items drawn from the original EAT-26 instrument, each targeting behavioral components associated with disordered eating. These items reflect behavioral patterns, such as food restriction (e.g., “I avoid eating when I am hungry”), purging behaviors (e.g., “I vomit after I have eaten”), and episodes of binge eating (e.g., “I have gone on eating binges when I feel that I may not be able to stop”). Responses were rated using a 6-point Likert scale, where elevated scores denoted greater engagement in disordered eating behaviors. For the purposes of this study, the nine items were organized into three distinct factors using an item parceling technique (see below). Internal consistency, as measured by Cronbach’s alpha, was 0.82 for the present sample, and the maximal reliability estimate for the EAT-26-B reached 0.87.

#### 2.3.3. Psychological Distress

The Depression Anxiety and Stress Scale, 21-item version (DASS-21; [25]) is a self-administered measure designed to assess psychological distress across three domains: depression, anxiety, and stress. Comprising 21 items, the instrument employs a 4-point Likert scale (ranging from 0 to 3), with higher scores indicating increased levels of distress. Prior studies have supported its convergent and discriminant validity ([33]). In this investigation, analyses were conducted using the original three-factor structure. Internal consistency for the current sample was high, with a Cronbach’s alpha of 0.93 and a maximal reliability estimate of 0.91.

#### 2.3.4. Difficulty in Emotion Regulation

The Difficulties in Emotion Regulation Scale (DERS; [19]) is a 36-item self-report instrument developed to assess deficits in emotion regulation. It comprises six theoretically derived subscales: (a) nonacceptance of emotional responses; (b) difficulties engaging in goal-directed behavior; (c) impulse control challenges; (d) lack of emotional awareness; (e) limited access to regulation strategies; and (f) lack of emotional clarity. Items are rated on a 5-point Likert scale ranging from 1 to 5, with elevated scores indicating greater difficulties in emotion regulation. Previous studies have demonstrated strong psychometric support for both the full scale and individual subscales ([19]). However, consistent with findings reported by [22] ([22]), the current study excluded the Awareness subscale, citing evidence that its removal does not compromise internal consistency and may reflect its distinctiveness as a construct. In the present sample, the remaining five-factor structure yielded a Cronbach’s alpha of 0.92 and a maximal reliability estimate of 0.91.

#### 2.3.5. Self-Concealment

The Self-Concealment Scale (SCS; [31]) is a 10-item self-report measure designed to assess the extent to which individuals habitually withhold personally distressing information. Each item is rated on a 5-point Likert scale ranging from 1 (strongly disagree) to 5 (strongly agree), with higher scores indicating greater levels of self-concealment. Previous research has demonstrated sound psychometric properties for the scale ([38]). Although originally conceptualized as a unidimensional measure, the present study employed an item parceling strategy to group the 10 items into three parcels for analytical purposes. Internal consistency was high, with a Cronbach’s alpha of 0.91 and a maximal reliability coefficient of 0.92.

### 2.4. Data Analyses

All statistical procedures were conducted using R software, version 4.1.2 ([50]). Descriptive statistics and Pearson correlation coefficients were computed to summarize and explore relationships among study variables. To assess normality assumptions, the absolute values of skewness and kurtosis were evaluated, with thresholds of 2 and 7, respectively, in accordance with guidelines by [57] ([57]).

Subsequently, following the two-step methodology outlined by [1] ([1]), both the measurement and structural models were evaluated sequentially. Model fit was assessed using three widely accepted indices: the root mean square error of approximation (RMSEA), the comparative fit index (CFI), and the Tucker–Lewis index (TLI). Consistent with established benchmarks, model fit was deemed adequate when RMSEA values were below 0.08 ([9]), and both CFI ([4]) and TLI ([5]) exceeded 0.90. After confirming a satisfactory fit for both models, path coefficients and overall model fit were compared between full and partial mediation models. A competing model strategy was employed, as previous literature has limited evidence to support whether the relationship between psychological distress and disordered eating behaviors was partially or fully mediated by difficulty in emotion regulation and self-concealment. By setting the partially mediated model as the research model and the fully mediated model as the competitive model, an attempt was made to search for a more accurate and parsimonious model. Structural Equation Modeling (SEM) analyses were conducted using the *lavaan* package in R ([54]).

To test the study’s hypotheses, bootstrapping procedures were employed to evaluate the statistical significance of indirect effects. This approach involved generating 5000 resampled datasets through random sampling with replacement from the original data. Indirect effects were considered statistically significant at the 0.05 level when the 95% confidence interval did not include zero ([7]).

As outlined earlier, item parceling was applied to the EAT-26-B and SCS scales, consistent with practices in SEM ([42]). This technique involves aggregating item scores, either by summation or averaging, to reduce the total number of measured variables and enhance model fit. Although parceling introduces potential bias, such effects are generally minimal and can be statistically adjusted. In alignment with the balancing approach proposed by [35] ([35]), items were grouped based on factor loading magnitudes to ensure relative equivalence within each parcel. To assess reliability, both composite reliability (e.g., Cronbach’s alpha) and maximal reliability ([52]) were calculated. Maximal reliability was included given its capacity to more accurately reflect the reliability of latent constructs as measured by their indicators. A threshold of 0.70 or greater is generally considered acceptable for maximal reliability estimates in empirical research. These calculations were conducted using the *semPlot* package in R ([13]).

## 3. Results

### 3.1. Demographics

As summarized in Table 1, the mean age of the retained sample was 19.5 years (*SD* = 1.6). The cohort was demographically diverse: 34.9% identified as Asian; 26.7% as Non-Hispanic White; 23.9% as Multiracial; 6.6% as Latinx; 3.5% as Hawaiian; 1.8% as Pacific Islander; 1.4% as Non-Hispanic Black; 0.6% as Other; and 0.5% as Native American. The racial and ethnic distribution of the final sample closely mirrored that of the broader student population at the host institution.

### 3.2. Measurement Model

Descriptive statistics, including means, standard deviations, skewness, kurtosis, and inter-variable correlations, were computed for all study variables and are presented in Table 2. The assumption of normality was met for all variables, as the observed skewness and kurtosis values fell within acceptable thresholds.

Prior to proceeding with structural model evaluation, a measurement model was evaluated for psychological distress, disordered eating behavior, difficulty in emotion regulation, and self-concealment. As shown in Table 3, the measurement model demonstrated a satisfactory fit to the data (*χ*^2^ = 232.263; *df* = 71; TLI = 0.968; CFI = 0.975; RMSEA = 0.056). The standardized factor loadings for the indicators were statistically significant at the 0.001 level and ranged from 0.83 to 0.92 for psychological distress, 0.67 to 0.91 for disordered eating behavior, 0.56 to 0.92 for emotion regulation difficulties, and 0.88 to 0.90 for self-concealment.

### 3.3. Structural Model

Since the above-mentioned measurement model adequately explained the latent variable, the structural model that explained the relationship between latent factors was examined. The result indicated that both partially mediated model (χ^2^ = 318.669; *df* = 111; TLI = 0.956; CFI = 0.968; RMSEA = 0.051) and fully mediated model (χ^2^ = 323.012; *df* = 112; TLI = 0.956; CFI = 0.967; RMSEA = 0.051) were good fit to the data. The chi-square difference test was employed to compare the model fit of two models, and a significant difference was identified (Δχ^2^ = 4.343; Δ*df* = 1; *p* < 0.05). If the chi-square difference is significant, the larger model with more freely estimated parameters fits the data better than the smaller model in which the parameters in question are fixed. Therefore, as shown in Figure 1, a partially mediated model was adopted as the final model.

### 3.4. Significance Test of Indirect Effects

A bootstrap procedure was performed to examine whether the indirect effect of psychological distress on disordered eating behavior via difficulty in emotion regulation and self-concealment was significant in the final model. We found the indirect effect of psychological distress on disordered eating behaviors through difficulty in emotion regulation and self-concealment (Table 4). The direct effect of psychological distress on disordered eating behaviors was also significant (β = 0.12, *p* = 0.05), which confirmed that the difficulty in emotion regulation and self-concealment partially mediates the effect of psychological distress. The direct effects of psychological distress on difficulty in emotion regulation and self-concealment were β = 0.65 (*p* < 0.01) and β = 0.45 (*p* < 0.01), respectively. The direct effect of difficulty in emotion regulation (β = 0.21, *p* < 0.01) and self-concealment (β = 0.15, *p* < 0.01) on disordered eating behaviors was also significant. The indirect effect between psychological distress and disordered eating behavior via difficulty in emotion regulation was β = 0.14 (*p* < 0.01), and the indirect effect via self-concealment was β = 0.07 (*p* < 0.01).

## 4. Discussion

Emerging adulthood is often marked by exposure to numerous life stressors ([3]), which may heighten vulnerability to disordered eating behaviors (e.g., [38]; [49]; [55]), and eating disorders continue to be more prevalent in women than men ([36]). This cross-sectional study investigated the relationship between psychological distress and disordered eating behaviors in a sample of emerging adult women, as well as the extent to which difficulty in emotion regulation and self-concealment mediated this association. Consistent with the study’s hypotheses, psychological distress was significantly correlated with disordered eating behavior. Further, both difficulty in emotion regulation and self-concealment were found to partially mediate the association between psychological distress and disordered eating behavior, suggesting their contributory roles in this linkage.

Conceptually, our results indicate that difficulty in emotion regulation and self-concealment function as interconnected yet distinct pathways that underlie the link between psychological distress and disordered eating behavior. These findings add to the growing research emphasizing the covert and maladaptive nature of disordered eating patterns (e.g., [18]; [37]; [39]). More specifically, the present study demonstrated that psychological distress was positively associated with disordered eating in emerging adult women, and this relationship was indirectly accounted for by both impaired emotion regulation and tendencies toward self-concealment. Clinically, there is a pressing need to improve treatment outcomes for eating disorders, as current response rates remain modest; approximately 50% for binge eating disorder ([26]) and 40–45% for bulimia nervosa ([46]). Our findings indicate that interventions targeting both emotion regulation difficulties and self-concealment may offer particular benefit for women in this age group.

Importantly, the present findings also indicate that the association between psychological distress and disordered eating behavior is not fully explained by difficulty in emotion regulation or self-concealment. This suggests the involvement of additional psychosocial factors, particularly among emerging adult women and potentially across other demographic groups. Such alternative contributing factors may include perfectionism, social anxiety, self-esteem challenges, and developmental stressors uniquely characteristic of emerging adulthood ([16]; [43]). Continued research in this domain holds meaningful clinical and theoretical relevance, offering potential avenues for refining interventions and prevention strategies aimed at disordered eating in this developmental group of women.

Furthermore, although it falls outside the scope of the current study, it is important to acknowledge the bidirectional nature of the association between psychological distress and disordered eating behavior. Disordered eating behaviors have been shown to intensify psychological distress over time, leading to heightened symptoms of anxiety, depression, and emotional dysregulation ([6]; [48]). This reciprocal dynamic suggests that disordered eating not only stems from psychological challenges but may also perpetuate them.

## 5. Limitations

Several limitations of the present study should be acknowledged. First, the sample consisted exclusively of emerging adult women enrolled at a public university in Hawaiʻi, which may constrain the generalizability of findings to other cultural and geographic contexts. Nonetheless, a notable strength of the study is the racial and ethnic diversity represented within the sample. Second, the findings are inherently tied to the specific instruments utilized ([23]), and alternative measures of psychological distress, disordered eating behavior, difficulty in emotion regulation, and self-concealment may yield different outcomes. Nevertheless, the selected instruments are widely recognized and frequently employed within psychological research. Additionally, by applying SEM, the study examined associations at the latent construct level, offering deeper insight into the underlying mechanisms. Third, given that this study employed a cross-sectional design, it was not possible to investigate the temporal or causal dynamics among key variables as proposed in the conceptual mediation framework. To clarify the roles of difficulty in emotion regulation and self-concealment within the interplay of psychological distress and disordered eating behavior, future investigations should adopt longitudinal methodologies or intervention-based approaches. Finally, a portion of the sample was collected after the onset of the COVID-19 pandemic. Considering the well-documented effects of the pandemic on psychological distress and eating behaviors (e.g., [10]; [51]), additional research is warranted to assess the generalizability of these findings.

## 6. Conclusions

The present investigation contributes meaningfully to the growing body of literature on disordered eating behaviors among emerging adult women and highlights important pathways linking psychological distress to disordered eating behaviors. By identifying difficulty in emotion regulation and self-concealment as partial mediators, the study offers insight into the psychological processes that may underlie maladaptive eating patterns in this population. These findings not only extend theoretical understanding but also underscore potential targets for prevention and clinical intervention efforts tailored to emerging adults.

## Figures and Tables

**Figure 1 behavsci-15-01259-f001:**
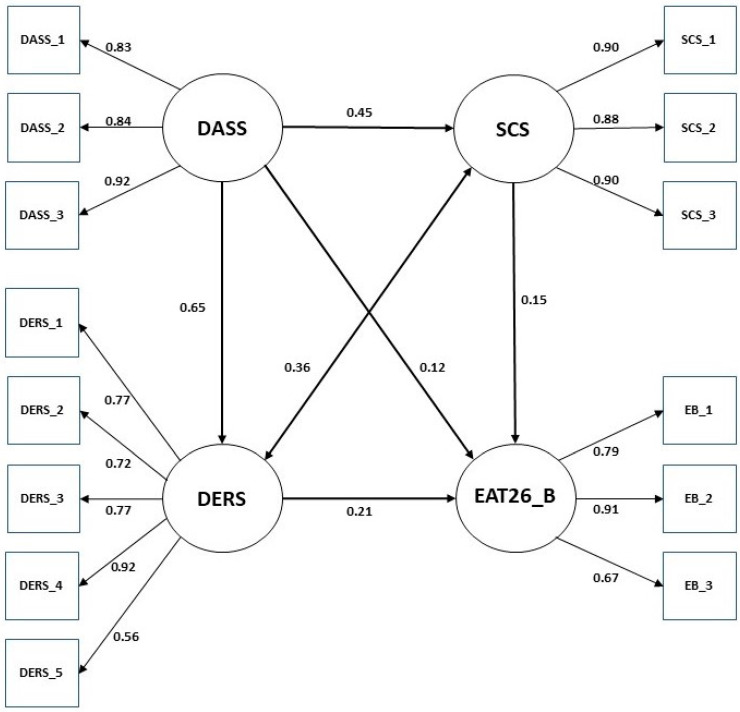
Partial mediation model between psychological distress and disordered eating behavior. Note: EAT26-B, the Eating Attitudes Test-26-Behavior Subscale; DASS, the Depression, Anxiety, and Stress Scale; DERS, the Difficulties in Emotion Regulation Scale; SCS, the Self Concealment Scale.

**Table 1 behavsci-15-01259-t001:** Demographic background.

		Total (*n* = 723)	
Characteristic		*M*	*SD*
Age		19.5	1.6
		Percent	*n*
Sexual Identity			
	Heterosexual	82.7	591
	Homosexual	2.1	15
	Bisexual	12.7	91
	Other	2.5	18
Race/Ethnicity			
	Native American	0.5	4
	Latinx	6.6	48
	Asian	35	253
	Pacific Islander	1.8	13
	White (non-Hispanic)	26.7	193
	Black (non-Hispanic)	1.4	10
	Hawaiian	3.5	25
	Other	0.6	4
	Biracial/Multicultural	23.9	173
Family Background			
	Poor	3.4	24
	Working Class	21.8	156
	Middle Class	47.6	341
	Upper Middle Class	24.9	178
	Wealthy	2.1	15

**Table 2 behavsci-15-01259-t002:** Means, standard deviation, and correlations of study variables.

Variables		1	2	3	4	5	6	7	8	9	10	11	12	13	14	15	16	17
1	EAT26_B1	-																
2	EAT26_B2	0.71 **	-															
3	EAT26_B3	0.56 **	0.64 **	-														
4	DASS_dep	0.24 **	0.27 **	0.12 **	-													
5	DASS_anx	0.25 **	0.26 **	0.11 **	0.69 **	-												
6	DASS_str	0.26 **	0.27 **	0.11 **	0.76 **	0.78 **	-											
7	DERS_non	0.30 **	0.31 **	0.18 **	0.44 **	0.40 **	0.44 **	-										
8	DERS_goal	0.16 **	0.18 **	0.06	0.45 **	0.40 **	0.46 **	0.52 **	-									
9	DERS_imp	0.28 **	0.29 **	0.09 *	0.43 **	0.44 **	0.48 **	0.54 **	0.61 **	-								
10	DERS_awa	0.16 **	0.19 **	0.04	0.19 **	0.13 **	0.17 **	0.26 **	0.11 **	0.23 **	-							
11	DERS_str	0.27 **	0.31 **	0.13 **	0.56 **	0.44 **	0.52 **	0.72 **	0.66 **	0.70 **	0.31 **	-						
12	DERS_cla	0.25 **	0.25 **	0.07	0.40 **	0.31 **	0.33 **	0.45 **	0.36 **	0.43 **	0.55 **	0.51 **	-					
13	SCS_f1	0.26 **	0.24 **	0.13 **	0.38 **	0.33 **	0.35 **	0.44 **	0.31 **	0.35 **	0.29 **	0.42 **	0.38 **	-				
14	SCS_f2	0.26 **	0.26 **	0.17 **	0.38 **	0.33 **	0.35 **	0.42 **	0.30 **	0.34 **	0.32 **	0.42 **	0.39 **	0.80 **	-			
15	SCS_f3	0.28 **	0.26 **	0.15 **	0.39 **	0.32 **	0.34 **	0.42 **	0.29 **	0.34 **	0.28 **	0.44 **	0.35 **	0.81 **	0.79 **	-		
16	Age	−0.07	−0.03	0.03	0.01	−0.05	0.03	−0.04	−0.01	0.01	−0.09 *	−0.03	−0.08 *	−0.07	−0.07	−0.08 *	-	
17	BMI	0.16 **	0.13 **	0.10 *	0.07	0.04	0.05	0.08 *	−0.03	0.02	0.04	0.07	0.07	0.02	0.03	0.06	0.13 **	-
	Mean	7.06	6.05	4.26	3.03	2.69	3.48	15.15	16.05	13.95	15.26	20.06	12.65	9.01	8.59	11.48	19.46	22.82
	*SD*	2.84	2.41	2.16	4.17	3.58	3.94	6.36	4.98	5.50	4.89	7.41	4.29	3.15	3.10	4.15	1.58	4.31
	Skewness	0.67	0.89	0.96	1.74	1.95	1.41	0.53	−0.01	0.70	0.28	0.41	0.40	−0.10	0.04	−0.09	1.25	1.70
	Kurtosis	0.40	0.77	0.56	2.88	4.43	1.83	−0.60	−0.96	−0.11	−0.42	−0.67	−0.49	−0.78	−0.79	−0.82	1.44	4.18

* *p* < 0.05, ** *p* < 0.01. Note: EAT26-B, the Eating Attitudes Test-26-Behavior Subscale; DASS, the Depression, Anxiety, and Stress Scale; DERS, the Difficulties in Emotion Regulation Scale; SCS, the Self Concealment Scale; BMI, Body Mass Index.

**Table 3 behavsci-15-01259-t003:** Confirmatory factor analysis of research model.

			β	*SE*	*z*	*p*
Disordered eating behaviors						
EAT26-B	→	Factor1	0.79			
EAT26-B	→	Factor2	0.91	0.05	20.30	<0.001
EAT26-B	→	Factor3	0.67	0.04	18.50	<0.001
Psychological distress						
DASS21	→	Depression	0.83			
DASS21	→	Anxiety	0.84	0.03	26.72	<0.001
DASS21	→	Stress	0.92	0.04	29.18	<0.001
Difficulty in emotion regulation						
DERS	→	Non-acceptance	0.77			
DERS	→	Goals	0.72	0.04	19.78	<0.001
DERS	→	Impulse	0.77	0.04	21.21	<0.001
DERS	→	Strategies	0.92	0.05	26.33	<0.001
DERS	→	Clarity	0.56	0.03	15.14	<0.001
Self-concealment						
SCS	→	Factor1	0.90			
SCS	→	Factor2	0.88	0.03	34.31	<0.001
SCS	→	Factor3	0.90	0.04	35.53	<0.001

Note: EAT26-B, the Eating Attitudes Test-26-Behavior Subscale; DASS, the Depression, Anxiety, and Stress Scale; DERS, the Difficulties in Emotion Regulation Scale; SCS, the Self Concealment Scale; β, standardized coefficient; *SE*, standard error; *z*, *z*-score; *p*, *p*-value.

**Table 4 behavsci-15-01259-t004:** Parameter estimate of partial mediation model.

					β (*SE*)	*z*	*p*	LLCI	ULCI
Direct effect									
DASS	→	DERS			0.65 (0.06)	14.87	<0.001		
DASS	→	SCS			0.45 (0.03)	11.13	<0.001		
DERS	→	EAT26-B			0.21 (0.03)	3.18	<0.001		
SCS	→	EAT26-B			0.15 (0.04)	2.94	<0.001		
DASS	→	EAT26-B			0.12 (0.04)	1.96	0.05		
Indirect effect									
DASS	→	DERS	→	EAT26-B	0.14 (0.03)	3.31	<0.001	0.04	0.14
DASS	→	SCS	→	EAT26-B	0.07 (0.02)	2.87	<0.001	0.15	0.27

Note: EAT26-B, the Eating Attitudes Test-26-Behavior Subscale; DASS, the Depression, Anxiety, and Stress Scale; DERS, the Difficulties in Emotion Regulation Scale; SCS, the Self Concealment Scale. β, standardized coefficient; *SE*, standard error; *z*, *z*-score; *p*, *p*-value; LLCI, Lower Limit of a Confidence Interval; ULCI, Upper Limit of a Confidence Interval.

## Data Availability

The datasets generated during and/or analyzed during the current study are available from the corresponding author on reasonable request.

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
