# Peer review of "Difficulty in Emotion Regulation and Self-Concealment as Mediators of the Link Between Psychological Distress and Disordered Eating Behavior in Emerging Adult Women"

_behavsci, 2025, doi:10.3390/bs15091259_

Round 1

Reviewer 1 Report

Comments and Suggestions for Authors

The manuscript title was “Difficulty in Emotion Regulation and Self-concealment as Mediators of the link between Psychological Distress and Disordered Eating Behavior in Emerging Adult Women”. The research was mainly about the relationship psychological distress and disordered eating behavior. The research was meanningful. The research was mainly expounds the content of one aspect which psychological distress is a major contributing factor of disordered eating behavior. In fact that the disordered eating behavior also has effect the psychological distress. I think this content should be discussed in detail also in the paper.

Reviewer 2 Report

Comments and Suggestions for Authors

I would like to thank the editors for the invitation to review this manuscript. The paper addresses an important and timely topic in the field of eating behavior and psychological processes among emerging adult women. Overall, the study is well written and makes a potentially meaningful contribution. At the same time, the manuscript would benefit from several modifications in order to strengthen its clarity, methodological transparency, and interpretive depth prior to publication.

Firstly, in the abstract, you mention: “Upon the completion of informed consent procedure, they voluntarily completed an online survey package that included the self-report measures of interest”. Could you add a few of the measures used.

The introduction is generally clear, but the study aims are stated twice, with overlapping wording: once in the opening paragraph (Line 39: “Extending this line of inquiry…”) and again at the end of the introduction (Line 98: Present study). The first mention reads almost like an explicit aim statement, which creates redundancy. You may wish to streamline this by keeping the full aim and hypotheses at the end of the introduction, reframing the earlier sentence as a transition from prior literature into the rationale for the present study, rather than presenting it as if it were already the aim.

In lines 112–114, the authors describe eligibility as correct response to four attention checks, self-identification as women, and age between 18 and 25 years. It is unclear whether any additional inclusion or exclusion criteria were applied. If none were considered, this should be explicitly stated. In addition, attention checks are usually presented as a data-quality measure rather than a formal eligibility criterion; clarification on this point would be appropriate.

In line 117, you present results, that should be placed in the result section, together with Table 1.

In line 126 and 352, can you write the number given for the Institutional Review Board Statement.

In line 204, you have referenced (Rosseel, 2012) twice.

Line 235: Please include the CFA description in your statistical analysis section, no need to describe it in the results, only mention the results of the CFA here.

Line 252-257: Could you explain this in the methodology and not in the results section “Competing model strategy was employed as previous literature has limited evidence to support whether the relationship between psychological distress and disordered eating behaviors was partially or fully mediated by difficulty in emotion regulation and self-concealment. More specifically, by setting the partially-mediated model as the research model and the fully-mediated model as the competitive model, it was attempted to search for a more accurate and parsimonious model”. ie. Give the information about partial mediation in the methods section.

Because data collection spanned March 2018 to December 2020, a portion of the sample was recruited during the COVID-19 pandemic. Given the well-documented impact of the pandemic on psychological distress and eating behaviors, it would strengthen the manuscript to acknowledge this contextual factor. Ideally, the authors could clarify whether data were examined separately for pre-COVID and during-COVID participants, or at minimum note this as a limitation in the discussion.

Reviewer 3 Report

Comments and Suggestions for Authors

Introduction:

Comment 1: Please remove the subtitles in the introduction, I consider them unnecessary

Comment 2: Sections 1.2, 1.2, and 1.3 can be written in a more concise and structured way to address the key concepts and definitions in the manuscript without so much extra information

Comment 3:

We are addressing health issues, so I see a lot of redundant phrases like the following:

Cognitive and behavioral theories…

these theories postulate that the positive…

Finally, theoretical models suggest that difficulty…

In this regard, I believe that the important thing would be to address the key concepts and reinforce the previous studies that support your study objective

Method:

Comment 4: The paragraph that says, "As summarized in Table 1, the mean age of the retained sample was 19.5 years…”, belongs to the results section, as well as table 1. Please place that information at the beginning of the results to know the characteristics of the sample, before addressing what they found in the survey

Comment 5: I would like to know more details and information about the survey conducted and please add it to the manuscript as supplementary material

Comment 6: Table 1

What does n mean?

Comment 7: Tables. Adds the meanings of b, SE, z and p

Comment 8: Discussion: The discussion is understandable; however I believe they should go into more depth by comparing their results with previous studies and the areas of opportunity or future directions that may arise from their manuscript

Comment 9: Discussion: Why do you consider your results relevant? Why does this type of study benefit women and why is it important to differentiate from a gender perspective?

Comment 10: Add a section specifically titled conclusion

Comment 11: Institutional Review Board Statement: Add the number and date of approval of the research protocol
